# Pursuit of chlorovirus genetic transformation and CRISPR/Cas9-mediated gene editing

Eric A. Noel [1,2¤], Donald P. Weeks[3], James L. Van Etten[1,4]*

1 Nebraska Center for Virology, University of Nebraska, Lincoln, Nebraska, United States of America,
2 School of Biological Sciences, University of Nebraska, Lincoln, Nebraska, United States of America,
3 Department of Biochemistry, University of Nebraska, Lincoln, Nebraska, United States of America,
4 Department of Plant Pathology, University of Nebraska, Lincoln, Nebraska, United States of America

¤ Current address: Innovative Genomics Institute, University of California, Berkeley, California, United States of America

* jvanetten1@unl.edu

**Data Availability Statement:** All relevant data are within the manuscript and its Supporting Information files.

**Funding:** This work was funded in part by National Science Foundation Grant 1736030 (JVE) and the National Science Foundation Graduate Research

## Abstract

Genetic and molecular modifications of the large dsDNA chloroviruses, with genomes of 290 to 370 kb, would expedite studies to elucidate the functions of both identified and unidentified virus-encoded proteins. These plaque-forming viruses replicate in certain unicellular, eukaryotic chlorella-like green algae. However, to date, only a few of these algal species and virtually none of their viruses have been genetically manipulated due to lack of practical methods for genetic transformation and genome editing. Attempts at using *Agrobacterium*-mediated transfection of chlorovirus host *Chlorella variabilis* NC64A with a specially-designed binary vector resulted in successful transgenic cell selection based on expression of a hygromycin-resistance gene, initial expression of a green fluorescence gene and demonstration of integration of *Agrobacterium* T-DNA. However, expression of the integrated genes was soon lost. To develop gene editing tools for modifying specific chlorovirus CA-4B genes using preassembled Cas9 protein-sgRNA ribonucleoproteins (RNPs), we tested multiple methods for delivery of Cas9/sgRNA RNP complexes into infected cells including cell wall-degrading enzymes, electroporation, silicon carbide (SiC) whiskers, and cell-penetrating peptides (CPPs). In one experiment two independent virus mutants were isolated from macerozyme-treated NC64A cells incubated with Cas9/sgRNA RNPs targeting virus CA-4B-encoded gene *034r*, which encodes a glycosyltransferase. Analysis of DNA sequences from the two mutant viruses showed highly targeted nucleotide sequence modifications in the *034r* gene of each virus that were fully consistent with Cas9/RNP-directed gene editing. However, in ten subsequent experiments, we were unable to duplicate these results and therefore unable to achieve a reliable system to genetically edit chloroviruses. Nonetheless, these observations provide strong initial suggestions that Cas9/RNPs may function to promote editing of the chlorovirus genome, and that further experimentation is warranted and worthwhile.

Fellowship Program Grant 250506019500 (EN)
(www.nsf.gov). The funders had no role in study
design, data collection and analysis, decision to
publish, or preparation of the manuscript.

**Competing interests:** The authors have declared
that no competing interests exist.

## Introduction

Research examining chloroviruses has provided many unexpected findings and concepts to
the scientific community over the past 40 years [1]. However, despite these major achieve-
ments, no transformation system has been developed that allows the genetic modification of
the large dsDNA viruses that infect certain unicellular *Chlorella*-like green microalgae. This
transformation bottleneck creates a significant handicap in exploration of chlorovirus
genomes that range from 290 to 370 kb and that encode many unidentified proteins. The abil-
ity to genetically engineer chloroviruses, in order to study and eventually manipulate their bio-
chemical pathways, would greatly enhance the utility of microalgae-chlorovirus counterparts
as scientifically and industrially important organisms. Considering that no reliable current
reverse genetics system exists for either *Chlorella variabilis* NC64A (hereafter NC64A) or
chloroviruses, we are equally limited in the capacity to either characterize gene function or
exploit unique virus-encoded proteins. With the advent of CRISPR technology and the ongo-
ing discovery of new giant viruses and their annotated genomes, we are armed with resources
that have yet to be married.

A significant barrier to genetic transformation of chloroviruses is the inaccessibility of its
host, NC64A, to DNA or protein uptake. Genetic engineering of microalgal strains is difficult
due to the great diversity of species with a variety of cell sizes, cell wall structures and composi-
tion and, likely, unique responses to foreign DNA [2]. Like plant cells, *Chlorella* cells are sur-
rounded by a rigid outer cell wall composed of polysaccharides with a variety of sugars as well
as lesser amounts of protein and lipid that presumably makes them more difficult to transform
[3]. DNA delivery can be challenging since DNA has to be transferred through the cell wall,
plasma membrane and nuclear membrane. Moreover, the cells must be able to survive the
chemical or mechanical treatments involved. Therefore, individualized protocols are needed
for specific strains and, thus, a broad range of genetic transformation methods must be
designed and tested.

Because microalgal cells are not able to take up exogenous DNA by nature, several genetic
techniques have been developed for this purpose. Among transformation methods for the
delivery of exogenous DNA, the most common techniques are electroporation, ballistic sys-
tems, agitation with glass beads and *Agrobacterium tumefaciens*-mediated transformation [4–
7]. Although, most of these techniques have been proven to work with great success in such
model algal strains as *Chlamydomonas reinhardtii*, *Phaeodactylum tricornutum*, *Scenedesmus*,
*Ankistrodesmus* and some *Chlorella* sp. [8, 9], there is a lack of efficient and stable transforma-
tion techniques that can be applied to a broader range of microalgae strains.

In earlier reports of successful genetic transformation of specific *Chlorella* species, various
methods were developed including the use of glass beads [10], *Agrobacterium tumefaciens*-
mediated transformation [11–13], PEG [14], protoplasting [15, 16], and electroporation [17–
19]. Electroporation has become the favored tool for DNA delivery and genetic transformation
of several microalgal species including *Chlamydomonas reinhardtii* [4], *Scenedesmus obliquus*
[20], and *Nannochloropsis* sp. [21]. However the protocol optimization is often challenging,
time-consuming and, most importantly, only proven in selected *Chlorella* sp. (*C. ellipsoidea*, *C.
vulgaris*, *C. minutissima*, *C. zofingiensis* and *C. pyrenoidosa*) [22]. Microalgal species display a
wide spectrum of resistance to transformation often based on differences in their ability to
take up and incorporate exogenous molecules into their genomes. Moreover, cell viability can
decrease rapidly when high voltages are needed and, likewise, increases in DNA fragment
lengths can also affect transformation efficiencies [23].

Less traditional methods of microalga-transformation are being explored with some suc-
cess. Karas *et al.* (2015) [24] and Diner *et al.* (2016) [25] showed that episomal plasmids

containing a yeast-derived centromeric sequence CEN6-ARSH4-HIS3 can be transferred by conjugation from *E. coli* strains to the diatoms *Thalassiosira pseudonana* and *Phaeodactylum tricornutum*. Recently, Munoz *et al.* (2019) [26] also reported an efficient and stable transformation of the green microalgae *Acutodesmus obliquus* and *Neochloris oleoabundans* by transferring exogenous DNA from *E. coli* via conjugation. Here, we initially attempted to use *Agrobacterium*-mediated transfection of chlorovirus host *Chlorella variabilis* NC64A with a specially-designed binary vector based on expression of a hygromycin-resistance gene. Prior to testing if NC64A cells could be genetically transformed using *A. tumefaciens*, we determined the levels of hygromycin needed to kill nontransformed NC64A cells and the levels of cefotaxime needed to rid cultures of *A. tumefaciens* following co-incubation of the algal and bacterial cells.

CRISPR/Cas systems have been widely used to manipulate the genomes of both freshwater and marine microalgae [27]. In particular, there are a number of reports in which Cas9/sgRNA-ribonucleoproteins-based approaches have been used for algal genome engineering. For example, the Cas9 protein and sgRNA are preassembled *in vitro*, and directly delivered to algal cells either via electroporation or by a biolistic method [28, 29]. In *C. reinhardtii*, Cas9/sgRNA RNP complexes were directly delivered to the cells by electroporation and created targeted mutations in multiple loci [28, 30]. RNP-based approaches have also been used to generate more robust strains of the industrial alga *Coccomyxa* as a biofuel cell factory [31].

In attempts to modify chlorovirus DNA, we tested previously described transformation protocols for other *Chlorella* sp. to deliver preassembled Cas9 protein/sgRNA RNPs inside NC64A cells prior to infection. As a gene to target for Cas9/sgRNA RNP modification, we chose a virus-encoded glycosyltransferase gene because we believed we could develop a screening scheme to select cells bearing mutations in the glycosylation pattern of the virus and because we wished to use such mutants to investigate the details of chlorovirus glycosylation. Specifically, we chose to target NC64A CA-4B virus-encoded gene, *034r*, which is a homolog of the prototype NC64A virus PBCV-1-encoded gene *a064r*, that encodes a highly characterized glycosyltransferase with three domains involved in protein glycosylation: domain 1 has a β-L-rhamnosyltransferase activity, domain 2 has an α-L-rhamnosyltransferase activity, and domain 3 is a methyltransferase (MT) that decorates the O-2 position in the terminal α-L-rhamnose unit [32]. The mutant selection scheme was based on the observation that CA-4B mutants can be selected by rabbit polyclonal antiserum derived from serologically distinct PBCV-1 mutants that have a mutation in gene *a064r* [33], that in turn produces truncated surface glycans. This antibody-based selection scheme, therefore, permits discrimination between wildtype viruses with native glycans decorating the major capsid protein (MCP) and viruses carrying an *a064r* gene mutation (likely caused by Cas9/sgRNA-directed gene editing) that produce a specific surface glycan variant.

The overall strategy to modify chlorovirus DNA involved testing a variety of transformation methods that could support the delivery of preassembled Cas9 protein-sgRNA RNP complexes to generate a targeted gene cleavage event in the CA-4B gene *034r*. Transformation methods that were investigated included protocols with cell wall-targeting enzymes, electroporation, silicon carbide (SiC) whiskers, and cell-penetrating peptides (CPPs). The RNA-directed selection of a specific 20–22 bp nucleotide sequence within the target gene *034r* by the Cas9/sgRNA complex allows the two nuclease domains of Cas9 to create a double stranded break (DSB) at a predetermined site within the gene of interest. Repair of the DSB by the error-prone nonhomologous end joining (NHEJ) DNA repair system can result in gene inactivation (i.e., gene knockout). In an even more powerful approach, replacement of the cleaved DNA segment with a closely related DNA fragment via homologous-directed recombination (HDR) can result in gene replacement (i.e., gene knockin) or nucleotide(s) substitution. Previously, it was

demonstrated that single-stranded oligodeoxynucleotides (ssODNs) could provide ~100-fold lower levels of nonhomologous integrations compared with double-stranded counterparts [34] while providing scarless genomic editing and reduced unwanted off-target cutting [35]. Thus, in an effort to increase efficiency and accuracy of targeted DNA editing and replacement in chloroviruses, we also attempted to co-deliver a DNA template in the form of ssODN with Cas9/RNPs that targets a 20 bp sequence within the first ~40 nucleotides of the *034r* domain 1 coding region. Our guide RNA was designed so that successful incorporation of a ssODN-mediated HDR event would remove a native MscI restriction site in *034r* while simultaneously introducing at a separate site a specific premature stop codon–a double event highly unlikely to occur by spontaneous gene mutation.

## Methods

### Alga growth conditions

NC64A cells were grown in Bold's basal medium (BBM) (3 mM $NaNO_3$, 170 μM $CaCl_2$, 304 μM $MgSO_4$, 431 μM $K_2HPO_4$, 1.3 mM $KH_2PO_4$, 428 μM NaCl, 12 μM $Na_2EDTA$, 2.2 μM $FeCl_3$, 1.2 μM $MnCl_2$, 220 nM $ZnSO_4$, 50 nM $CoCl_2$, 99 nM $Na_2MoO_4$, 6.4 μM $CuSO_4$, 184 μM $H_3BO_3$) modified by the addition of 0.5% sucrose and 0.1% peptone (MBBM) [36]. All experiments were performed with cells grown to early log phase ($4–7 \times 10^6$ cells/mL). Cell cultures were shaken (200 rpm) at 26°C under continuous light.

### *Agrobacterium* strain and vectors

The binary vector pCAMBIA1304 containing a *gfp-gusA* fusion reporter, gene a*064r* from a chlorovirus PBCV-1 antigenic mutant referred to as EPA-1, and a selectable marker for hygromycin B resistance driven by the CaMV 35S promoter were used for transformation. The binary vector was mobilized into *A. tumefaciens* strain LBA4404 by using the Biorad Gene Pulser Xcell electroporator (Hercules, CA) according to manufacturer's protocol. Transformed cells were aliquoted and maintained at -80°C in 25% (v/v) glycerol.

### Antibiotic sensitivity test

The sensitivity of *A. tumefaciens* towards the antibiotic cefotaxime was tested by inoculating 200 μL of *Agrobacterium* culture ($OD_{600}$ = 1.0) in 5 mL LB broth supplemented with varying concentrations of cefotaxime (0, 50, 100, 150, 200, 300, 400 and 500 mg/L) and the growth of *Agrobacterium* in each concentration was measured spectrophotometrically at $OD_{600}$ after 2 days. The effect of the antibiotic cefotaxime on the viability of NC64A was accessed by plating a serially-diluted microalgae culture on solid MBBM supplemented with different concentrations of cefotaxime (0, 100, 200, 300, 400 and 500 mg/L). The agar plates were incubated in the dark for 2 days at 25°C before exposure to light and the number of surviving colonies from the dilution that produced less than 100 colonies was counted in duplicates after 2 weeks. To determine the minimum inhibitory concentration of hygromycin B, 1 x $10^6$ NC64A cells were plated on solid MBBM medium supplemented with 500 mg/L cefotaxime and varying concentrations of hygromycin B (6, 8, 10, 12, 14, 16, 18, 20, 23 and 26 mg/L). Each treatment was tested in triplicate. The agar plates were incubated for 2 days in the dark at 25°C before exposure to light and the number of surviving colonies was counted after 20 days.

### NC64A transformation using *Agrobacterium*

A general transformation procedure for NC64A was established based on work done by Kumar *et al.* (2004) [5] on transformation of *C. reinhardtii* with some adjustment as described

here. Single colonies of *Agrobacterium* initiated from a frozen stock were used to inoculate 10 mL of LB supplemented with 5 mM glucose, 100 mg/L streptomycin and 50 mg/L kanamycin and grown overnight in a rotary shaker at 27˚C with shaking at 200 rpm in the dark. Five mL of this overnight culture was used to inoculate 50 mL of the same medium and it was grown in the dark at 27˚C with shaking at 200 rpm until $OD_{600}$ = 0.8–1.2. The bacterial culture was harvested by centrifugation and washed once with induction medium (MBBM plus 150 μM acetosyringone, pH 5.6) and diluted to a final density of $OD_{600}$ = 0.5. Prior to co-cultivation, a total of 5 x $10^6$ NC64A cells from a log-phase culture ($OD_{600}$ = 0.5–1.0) were pre-cultured for 5 days on MBBM agarose plates at 25˚C and harvested with induction medium on the day of co-cultivation. The algal cell pellet was mixed with 200 μL of the bacterial suspension and plated on induction medium solidified with 1.2% (w/v) bacto-agar. Co-cultivation was performed for 3 days at 25˚C in the dark. Following co-cultivation, cells were harvested with MBBM supplemented with 500 mg/L cefotaxime in a total volume of 7 mL and incubated in the dark at 25˚C for 2 days to eliminate *Agrobacterium*. Cells were plated on selective media containing 20 mg/L hygromycin B and 500 mg/L cefotaxime and incubated at 25˚C in the dark for 2 days before exposure to light. Resistant colonies were propagated on non-selective media and utilized for PCR analysis. Assays for detection of contaminating *Agrobacterium* were performed by growing cells on LB agar plates for at least 7 days at 25˚C in the dark.

## Cell wall-degrading enzymes

Cellulase, chitinase, chitosanase (25.9 U/mL), drieselase, β-glucosidase, β-glucuronidase (140 U/mL), hyaluronidase, laminarinase, lysozyme, lyticase, macerozyme, pectinase (3,000 U/mL), pectolyase, sulphatase (3.37 mg/mL), and trypsin were purchased from Sigma Aldrich (St. Louis, MO). Zymolyase (10 mg/mL) was purchased from ZymoResearch (Irvine, CA). Macerozyme was purchased from RPI corp (Mt Prospect, IL). Stock concentrations of enzymes were 20 mg/mL unless otherwise noted.

## *In vitro* Cas9/sgRNA-directed DNA cleavage assay

Production of chloroviruses PBCV-1, CA-4B and DNA isolation were performed as described [37]. An 835 bp CA-4B DNA target for Cas9/sgRNA cleavage was PCR-amplified and purified using 2% agarose gel electrophoresis (DNA sequences for primers used for DNA amplification are provided in the legend of S1 Fig). Purified *Streptococcus pyogenes* strain Cas9 (SpyCas9) (200 nM) was preincubated with sgRNA (600 nM) in cleavage buffer [1× NEBuffer 3 (New England Biolabs, Ipswich, MA), 10 mM DTT, 10 mM $CaCl_2$] at 37˚C for 15 min. Target DNA (20 nM) was added to a final volume of 20 μL. Reactions were incubated at 37˚C for 1 h. DNA in cleavage reactions was purified by using a MinElute PCR Purification Kit (Qiagen), resolved by size on a 2% agarose gel and imaged on Gel Doc XR+ and ChemiDoc XRS+ systems (Biorad).

## ssODN construct

We used tandem co-delivery of a DNA template in the form of ssODN with Cas9/sgRNA RNPs in attempts to achieve HDR. The ssODN was 80 nt long, designed with homology arms extending 40 nt upstream and downstream of the sgRNA target in CA-4B *034r*, respectively. The ssODN contained two critical elements that permitted selection when incorporated into a mutant virus: (1) a point mutation (T to A) that converts a native MscI restriction site `TGGCCA` to `AGGCCA`, and (2) a separate site nucleotide substitution (C to A) that introduces a premature stop codon (TAA). A successful HDR event would remove the MscI restriction site and allow initial verification of HDR-mediated insertion by simply treating PCR amplicons

with MscI and showing loss of the restriction site near the site of Cas9/RNP-mediated DNA cleavage. Amplicons were subsequently sequenced to determine if Cas9/sgRNA- and ssODN-directed nucleotide replacement had occurred. The simultaneous presence of a premature stop codon would eliminate translation of the glycosyltransferase enzyme and result in a shortened glycan at the MCP surface of newly formed viruses. Such mutants can be discriminated from wild-type viruses in the antibody-based selection scheme described below.

## CA-4B mutant selection using mutant glycan-specific antibodies

NC64A cells were infected with CA-4B (MOI 5) and incubated for 12 h (detailed below). Site-directed mutant viruses were selected with anti-Rabbit IgG magnetic beads (RayBiotech) according to the manufacturer's protocol. In brief, rabbit polyclonal antiserum prepared against wild-type PBCV-1 (which also binds CA-4B) and collected during previous antibody studies [33], was added to a virally-induced cell lysate (see details below) and allowed to bind wild-type virus for 1 h. Goat anti-Rabbit IgG magnetic beads were incubated with the rabbit antibody solution for 30 min and then separated using magnets. The unbound viruses were collected and subsequently incubated for 1 h with rabbit polyclonal antiserum derived from serologically distinct PBCV-1 mutants that have a mutation in gene *a064r* (homologous to CA-4B gene *034r*) that produces a specific truncated surface glycan. Goat anti-Rabbit IgG magnetic beads were incubated again with the rabbit antibody solution for 30 min and then separated with magnets. After the unbound particulates were washed from the beads, the bound mutant antibodies were eluted from the beads using the elution buffer. The beads were then magnetically separated from the eluted solution. The eluted antibodies coupled to mutant viruses were removed manually and the viruses were then plaque assayed. Individual plaques were selected, and target DNA regions were PCR amplified with custom oligonucleotide primers (sense 5′-GCGGTGTTCTCTAAATTACC-3′; antisense 5′-CCAGTTGCTACCATCTC C-3′; IDT, Coralville, IA). The PCR products were size-verified using agarose gel electrophoresis, eluted from the gel and sequenced using the Sanger method (GENEWIZ, South San Francisco, CA).

## Assay of enzyme inhibited cell growth

To assay growth inhibition due to enzymatic activity, 200 μL of algal cells normalized to an $OD_{750}$ of 1.0 were mixed with 4 mL of media containing 1.5% agar, which was at 42˚C and poured on a petri plate containing 15 g/L agar. Once hardened, 10 μL of enzyme stock was then spotted on the top agar and plates were incubated in the light at 26˚C for 5 days. As a negative control for enzymatic activity, enzymes were heat denatured at 100˚C for 10 min and spotted onto plates.

## NC64A transformation following enzymatic digestion of cell walls

For cell wall digestion, one mL of cell culture of NC64A ($5 \times 10^6$ cells) was centrifuged at 8,000 rpm for 5 min and the pellet was resuspended in the same volume of MBBM in the presence of cell wall-degrading enzymes or MBBM only. The culture was incubated for 24 h at 25˚C in continuous light. NC64A cells were centrifuged and resuspended in a solution (0.8 M NaCl and 0.05 M $CaCl_2$). Prior to NC64A cells being tested for their ability to be transformed with Cas9/sgRNA RNPs, all treated cells were infected with virus and analyzed by plaque assay to confirm treated cells retained their permissive qualities required for virus attachment and infection. To generate target-specific site-directed mutants using RNP complexes in NC64A, enzyme-treated and nontreated cells were transformed with Cas9 (200 μg) premixed with *034r*-targeting sgRNA (140 μg). Cas9 in storage buffer (20 mM HEPES pH 7.5, 150 mM KCl, 1

mM DTT, and 10% glycerol) was mixed with sgRNA dissolved in nuclease-free water and incubated for 10 min at room temperature. The mixture of cells and Cas9/sgRNA RNPs was incubated for 15 min at room temperature. Cas9/sgRNA-transformed, enzyme-treated NC64A cells were then infected with CA-4B and incubated overnight. Virus mutants were selected using the same antibody selection assay previously described. DNA from virus plaques were analyzed by PCR and sequenced to detect indel mutations. In experiments testing for ssODN-mediated HDR, *034r* gene PCR amplicons were tested for sensitivity to cleavage by MscI prior to DNA sequencing.

## Flow cytometry

Cell permeability assays were performed on the BD FACSAria cell sorter using ATTO[TM] 550 labeled Alt-R[TM] Cas9 tracrRNA from IDT (Coralville, IA). The 67mer Alt-R[TM] CRISPR-Cas9 tracrRNA has an ATTO[TM] 550 fluorescent dye attached to the 5' end. The fluorescent dye allows for an optical analysis of transfected cells and cell sorting by FACS. During data acquisition, algal cells were positively defined by their chlorophyll autofluorescence. A minimum chlorophyll autofluorescence was set to eliminate potential false positives from bacteria and debris present in the culture. One μL of ATTO[TM] 550-labeled tracrRNA (200 uM) was added, incubated for 2 min, and loaded on the BD FACSAria where 20,000 cells were imaged. Samples were excited using a 488 nm laser and 660–740 nm (chlorophyll) and 480–560 nm (ATTO) emission data as well as bright field image data were collected. Populations were gated for in-focus cells and analyzed for permeability to labeled tracrRNA.

## NC64A transformation using SiC whiskers

For each transformation, NC64A cells were concentrated and resuspended in 800 μL fresh MBBM ($5 \times 10^7$ cells) in a 1.5 mL microfuge tube and mixed with 50 mg sterile SiC whiskers and preassembled Cas9/sgRNA RNP (armed with the protospacer sequence targeting domain 1 of CA-4B *034r*) in the presence and absence of the ssODN. Samples were agitated by a vortex mixer at top speed for 2 min, stopping briefly every 10 s. Immediately after vortexing, samples were diluted with 200 μL PEG (20% PEG 8000) reaching a final volume of 1 mL, and infected with CA-4B virus as described above. Following overnight infection, virus mutants were selected using the same antibody selection assay previously described. DNA from virus plaques were analyzed by PCR and sequenced to detect indel mutations caused by NHEJ and/or other error-prone DNA repair mechanisms, while *034r* gene PCR amplicons were also incubated with MscI to screen for ssODN-mediated HDR.

## NC64A transformation using electroporation

Purified Cas9 (100 μg) was preincubated at a 1:3 molar ratio with the *034r*-targeting sgRNA at 37°C for 15 min to form RNP complexes. For transfection, 250 μL cell culture ($2.5 \times 10^8$ cells/mL) supplemented with sucrose (40 mM) were mixed with preincubated RNPs, and 150 nM ssODNs. Cells were electroporated in 2-mm cuvettes (600 V, 50 μF, 200 Ω) by using Gene Pulser Xcell (Biorad). Immediately after electroporation, 800 μL of MBBM with 40 mM sucrose was added to the sample and the cells were infected with CA-4B (MOI 5). Following overnight infection, virus mutants were selected using the same antibody selection assay previously described. Virus plaques were analyzed by PCR and sequenced to detect indel mutations, while *034r* gene PCR amplicons were also incubated with MscI to screen for ssODN-mediated HDR.

## Cell-penetrating peptides

Peptides were provided by Dr. Heriberto Cerutti (UNL) and Keiji Numata (RIKEN): (BP100)$_2$K$_8$ (KKLFKKILKYLKKLFKKILKYLKKKKKKKK, theoretical pI/Mw: 10.75/3851.13 Da) and BP100(KH)$_9$ (KKLFKKILKYLKHKHKHKHKHKHKHKHKH, theoretical pI/Mw: 10.81/3809.71 Da) [38]. Peptide/RNP complexes were prepared by adding different amounts of each peptide to the Cas9/sgRNA RNP mixture at various ratios (0.1, 0.5, 1, 2, 5, 10, and 20). The solution was thoroughly mixed by repeated pipetting and allowed to stabilize for 30 min at 25˚C.

## NC64A transformation using CPPs

Referring to a previous study [30], 10 µg of Cas9 and 12 µg of sgRNA were incubated at room temperature for 15 min. A prepared cell sample of 100 µl at $3 \times 10^8$ cells/mL was added to the incubated RNP, ssODNs, and gently mixed. Tested independently, (BP100)$_2$K$_8$ or BP100 (KH)$_9$ [38] was added to the sample and mixed immediately. After incubation of the cells mixed with the RNP and CPP for 30 min at 25˚C, trypsin was added, and the mixture was incubated for 15 min at 37˚C. The sample was washed by MBBM media and transferred to 10 mL of MBBM media and incubated for 16 h under dim light without shaking as a "recovery" step. Following incubation, cells were infected with CA-4B (MOI 5) and incubated overnight (24 h). Following overnight infection, virus mutants were selected using the same antibody selection assay previously described. DNA from virus plaques were analyzed by PCR and sequenced to detect indel mutations, while *034r* gene PCR amplicons were also incubated with MscI to screen for ssODN-mediated HDR.

# Results

## *Agrobacterium tumefaciens* transformation of NC64A

Prior to testing the ability of *Agrobacterium tumefaciens* to genetically transform NC64A, experiments were performed to determine the concentrations of cefotaxime needed to rid cultures of NC64A of *Agrobacterium* following algal/bacterial co-cultivations (i.e., transformation reactions) and the level of hygromycin needed to kill all NC64A cells not integrating *Agrobacterium* T-DNA containing the hygromycin-resistance gene into their genome. Growth of *A. tumefaciens* was inhibited at a cefotaxime concentration of 100 mg/L (Fig 1A) whereas the growth of NC64A was found to be uninhibited in cefotaxime-supplemented media at the same concentration up to at least 1000 mg/L (Fig 1B). Thus, 500 mg/L of cefotaxime was selected for all experiments to ensure thorough elimination of *Agrobacterium* post-transformation. The lowest concentration of hygromycin B which completely inhibited the growth of NC64A was 20 mg/L (Fig 1C), and this concentration was used for subsequent selection of transformants. To verify successful electroporation of the pCAMBIA1304 binary vector carrying a kanamycin resistance gene (see below) into the *A. tumefaciens* strain to be used for transforming NC64A cells, colony PCR was performed.

## PCR-confirmed *Agrobacterium*-mediated transformation of NC64A cells

To determine if *A. tumefaciens* could be used to transfer T-DNA containing foreign genes into the genome of NC64A cells, we mixed the algal cells with *A. tumefaciens* strain LBA4404 harboring the binary vector pCAMBIA1304 containing the *gfp-gusA* fusion reporter, a hygromycin phosphotransferase (*hpt*) selectable marker driven by the CaMV35S promoter, and the mutant a*064r* glycosyltransferase *gene* from a PBCV-1 antigenic mutant referred to as EPA-1 (EPA1-*a064r*). The mutant gene contains a missense mutation (C→T) at nucleotide position

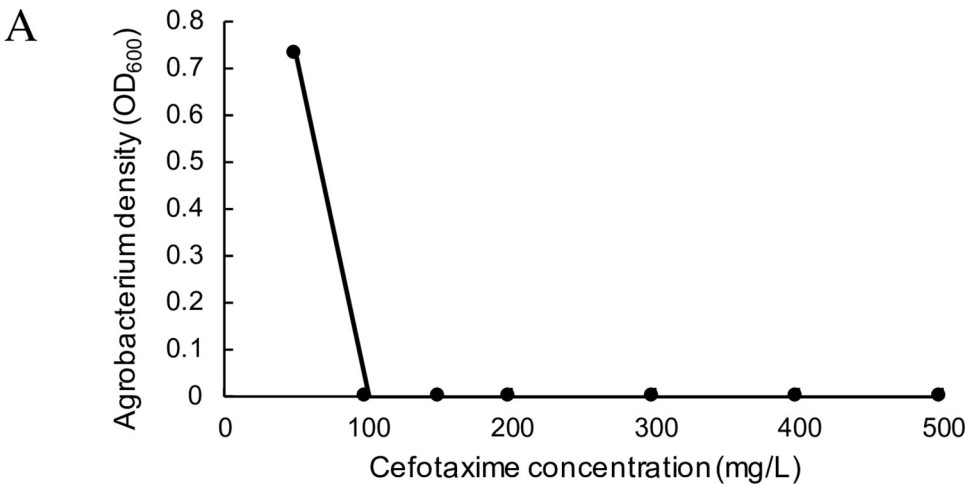

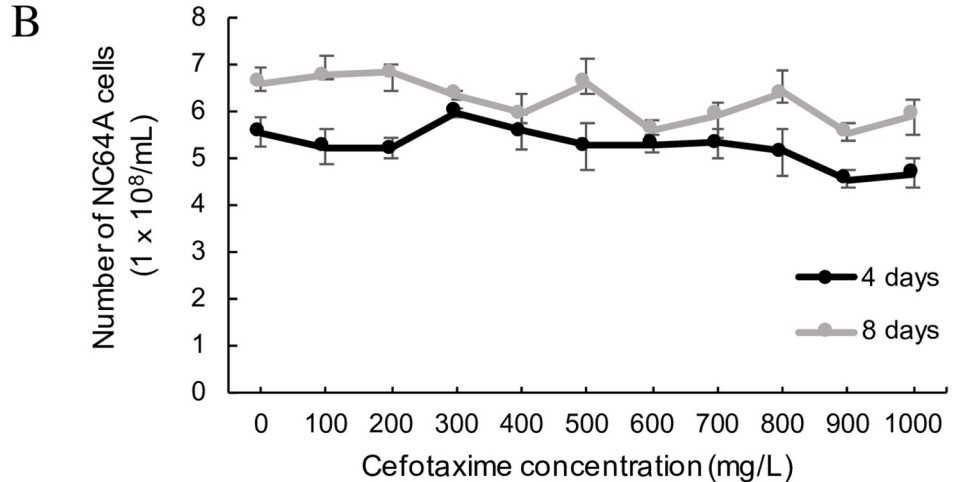

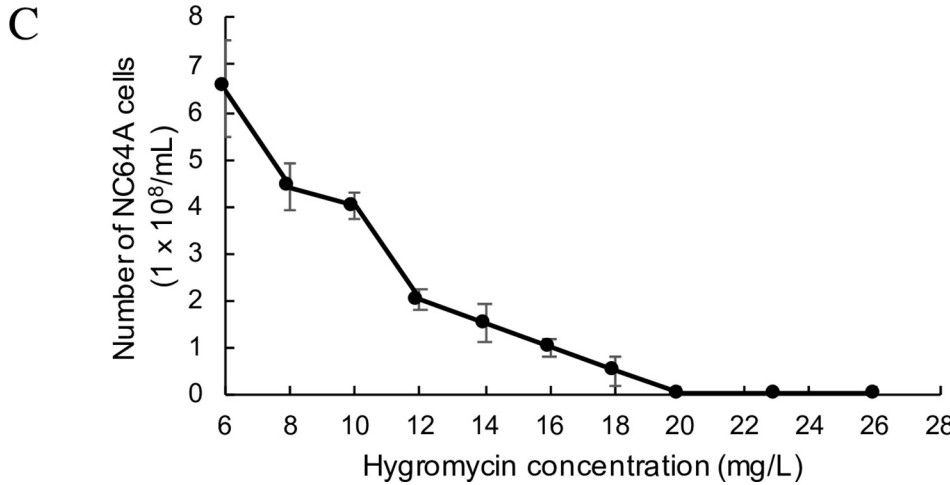

**Fig 1. The effects of antibiotics on *Agrobacterium* and NC64A.** (A) The effect of cefotaxime on *Agrobacterium* viability. The effects of (B) cefotaxime and (C) hygromycin on NC64A viability.

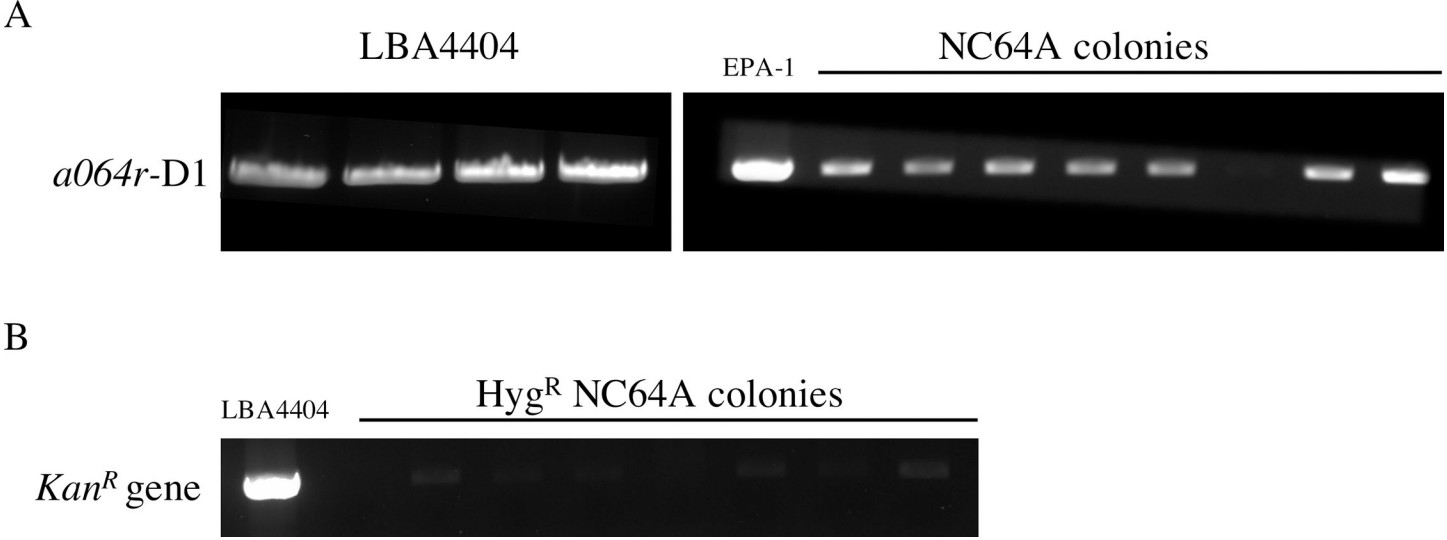

**Fig 2. PCR analysis of putative NC64A transformants transformed with recombinant pCAMBIA1304 *034r* domain 1.** (A) Amplification of the 633 bp fragment of the *a064r* domain 1 gene fragment from four LBA4404 colonies post-electroporation, and eight NC64A colonies after *Agrobacterium*-mediated transformation. (B) Amplification of the *Kan^R* gene from LBA4404 (positive control) but not in the seven putative NC64A transgenic lines.

236 resulting in an amino acid substitution (S79L) that causes production of a truncated glycan at the surface of the MCP [39]. NC64A transformation was assessed by monitoring transient *β*-glucuronidase (GUS) and *gfp* expression 2 days after algal and bacterial co-incubation.

From among a few hundred colonies containing putative transformants that appeared within 20 days on selection media (containing 20 mg/L hygromycin and 500 mg/L cefotaxime) a total of eight hygromycin-resistant single colonies were randomly selected and grown in liquid media (containing 500 mg/L cefotaxime but lacking hygromycin) before the DNA was extracted and used in PCR analysis. Amplification with EPA1-*a064r* domain 1 (D1) gene-specific primers successfully detected the 633 bp a*064r* gene fragment from seven putative transgenic lines (Fig 2A), which represents 87.5% of the total number of screened colonies. To rule out *Agrobacterium* contamination, amplification with Kan$^R$ gene-specific primers only produced the expected 795 bp Kan$^R$ gene fragments in LBA4404 (positive control) but not in the seven putative NC64A transgenic lines (Fig 2B). The presence and absence of these gene-specific fragments in seven putative transgenic lines with no amplification product detected in non-infected wild-type NC64A cells indicated the successful mobilization of both the hygromycin marker and the *gfp-gusA* reporter gene as a single T-DNA unit flanked by the left and right borders. The identity of the PCR amplicons derived from transgenic NC64A colonies were further confirmed to be positive by DNA sequencing where sequence alignment showed 99 and 98% identity to the *hpt* and *gfp-gusA* genes in pCAMBIA1304 respectively (Gene Bank accession no. AF234300).

### Chitinase, laminarinase, lysozyme, lyticase, macerozyme, and pectinase enzymes inhibit growth of NC64A

Inhibition of cell growth by an active preparation of cell wall-degrading enzyme suggests the enzyme is either degrading the cell wall during its construction or that the enzyme interferes with precursor generation prior to precursor assembly into the cell wall. From a variety of enzymes tested (Table 1), chitinase, laminarinase, lysozyme, lyticase, macerozyme, and pectinase completely inhibited growth of NC64A as judged by the size of the zone of inhibition

**Table 1. Growth inhibition of *Chlorella variabilis* NC64A by a variety of enzymes.**

| Enzyme | NC64A |
|---|---|
| Cellulase | – |
| Chitinase | + + + |
| Chitosanase | + |
| Driselase | – |
| *β*-Glucosidase | – |
| *β*-Glucuronidase | – |
| Hyaluronidase | – |
| Laminarinase | + + + |
| Lysozyme | + + + |
| Lyticase | + + + |
| Macerozyme | + + + |
| Pectinase | + + + |
| Pectolyase | – |
| Sulphatase | + |
| Trypsin | – |
| Zymolyase | – |

+++, complete growth inhibition; ++ moderate growth inhibition; +, slight growth inhibition; –, no growth inhibition.

around the spotted area. By comparison, chitosanase and sulphatase caused minor growth inhibition. Heat denatured enzymes did not inhibit growth in the assay, indicating the growth inhibition was due to the enzymatic activity alone and not from other potentially toxic components in the enzyme preparations.

## Preassembled RNPs with fluorescently labeled tracrRNA enter enzyme-treated NC64A cells

Fluorescently labeled Alt-R™ Cas9 tracrRNA was used to evaluate cell permeability and visualized using flow cytometer (BD FACSAria). The resulting data were analyzed by setting a minimum ATTO™ 550 fluorescent intensity threshold, such that the majority of untreated cells (Fig 3A) had fluorescence intensity lower than the threshold. Cells with a fluorescent intensity above this threshold were considered permeable to the tracrRNA dye as indicated by cells pretreated with macerozyme (Fig 3B), or vortexed with SiC whiskers or electroporated prior to fluorescence-activated cell sorting (FACS) (Fig 3C and 3D, respectively). Cells contained 5.6%, 12.4%, and 18.6% of total sample fluorescence when treated with either macerozyme, vortexed with SiC whiskers, or electroporated, respectively. In contrast, cells not treated with enzymes contained less than 0.1% of total sample fluorescence. In a series of experiments, we observed no significant difference between two types of treatment with CPPs: (1) pre-incubation of a mixture of RNP and CPP, and (2) addition of CPP and RNP separately to the cells.

## Design, production, and testing of a sgRNA targeting a specific site in the CA-4B virus *034r* gene

A sgRNA was designed to cleave the CA-4B virus in the first domain of the glycosyltransferase gene *034r* at 27 nucleotides into the coding region (Fig 4). To ensure this molecule was active and accurate in cleaving target DNA when combined *in vitro* with *Streptococcus pyogenes* strain Cas9 (SpyCas9), the appropriate sgRNA fragment (denoted in Fig 4) was synthesized

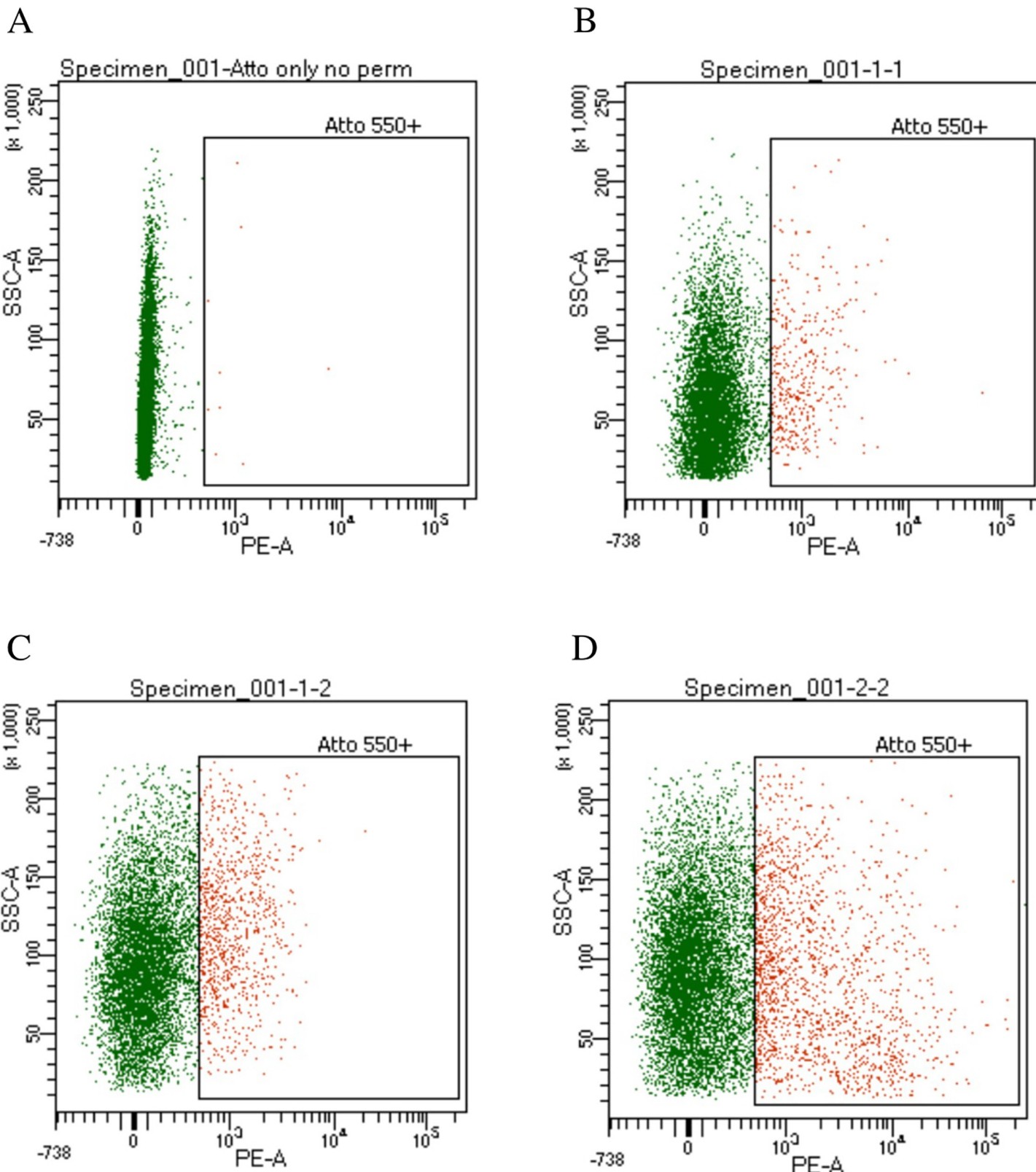

**Fig 3. FACS dot plot analysis of fluorescently labeled NC64A cells.** NC64A cells were visualized by intracellular incorporation of fluorescently labeled tracrRNA (ATTO^{TM} 550), (A) without enzyme treatment (0.1% fluorescence), or prior to FACS were (B) macerozyme digested (5.6% fluorescence), (C) vortexed with SiC whiskers

(12.4% fluorescence), or (D) electroporated (18.6% fluorescence). Cells were illuminated with a white LED for bright field and a 488 nm laser for fluorescence. Images were collected in the bright field and 480–560 nm emission (ATTO™ 550 fluorescence) channels. Cell populations with a high fluorescent intensity were gated for in-focus cells and analyzed for permeability.

and tested for its ability to cut a PCR-amplified 835 bp fragment of the CA-4B virus first domain. The data of S1 Fig confirmed the ability of the *in vitro*-assembled Cas9/sgRNA RNP complex to accurately cleave the CA-4B viral DNA fragment into the expected 600 bp and 235 bp fragments.

## Antibody-based selection and recovery of Cas9/sgRNA RNP-induced chlorovirus mutants from macerozyme-treated NC64A cells

We have shown previously that CA-4B is recognized by antibodies to PBCV-1 and by antibodies to certain PBCV-1 antigenic variants (mutants) making such antibodies useful to isolate wild-type and mutant versions of CA-4B virus [40]. In addition, studies involving spontaneous mutant viruses have demonstrated previously that disruption of PBCV-1 *a064r* leads to viruses with an altered glycan attached to the MCP and that these mutants are antigenically different from wild type PBCV-1 [41]. In additional studies [33, 39], polyclonal antisera were prepared against these spontaneous mutants. This allowed us to screen for viruses containing a disrupted *034r* gene using antibody that specifically recognized viruses with a gene-inactivating mutation in the homologous *a064r* gene.

Macerozyme-treated NC64A cells were mixed with Cas9/sgRNA RNP complexes targeting *034r* and subsequently infected with CA-4B overnight. Following incubation of viruses recovered from ruptured cells with the serologically distinct antigenic antibody, the putative mutant precipitate was eluted, and plaque assayed (Fig 5). The presence of mutations in the resulting plaques were subsequently verified by DNA sequencing (in triplicate DNA sequencing reactions) of domain 1 from *034r* using appropriate PCR primers. Two independent plaques showed separate modifications at the Cas9/sgRNA cleavage site [one nucleotide deletion in DNA from one plaque and one nucleotide substitution (i.e., on nucleotide deletion and one

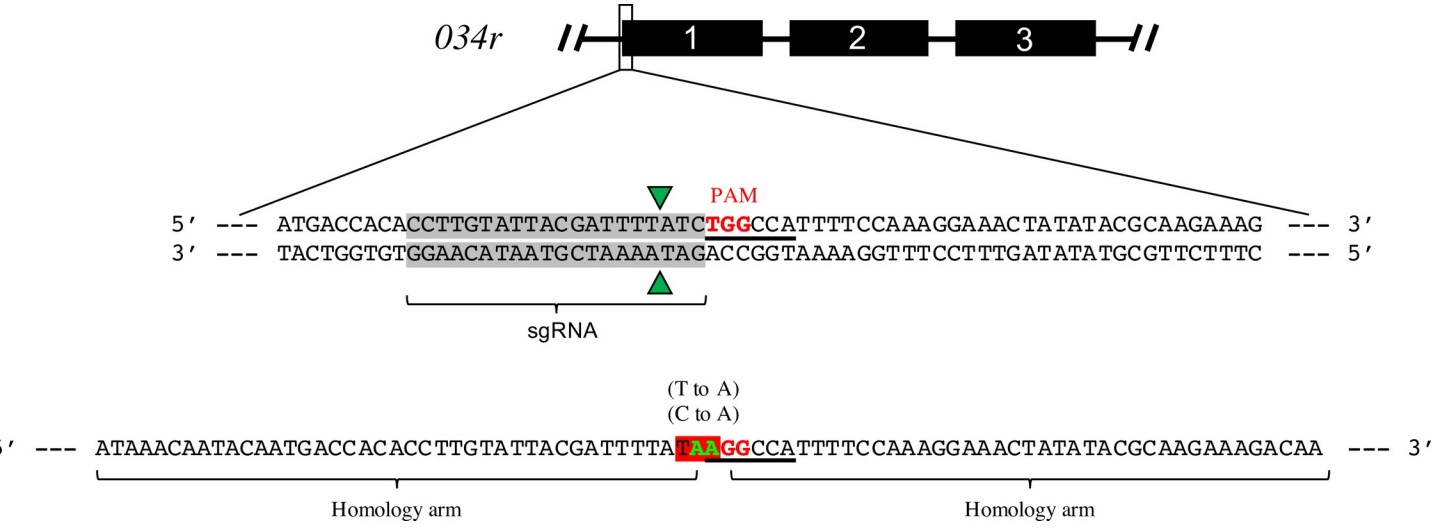

**Fig 4. sgRNA and ssODN designs for targeting CA-4B glycosyltransferase-gene *034r*.** Top: Cartoon rendering of CA-4B glycosyltransferase-gene *034r* composed of three domains. Middle: DNA sequence of the initial coding region of the *034r* gene with the target site for Cas9/sgRNA binding, cleavage and editing highlighted in gray. Red TGG, PAM site; green triangles, Cas9 cleavage sites; underlined, MscI restriction site. Bottom: Design of ssODN to replace the dinucleotide, CT, with AA (green) in the target gene. The homology arms specific to the gene target are flanking the nucleotides designed to be changed (green font). Shaded red, newly created stop codon.

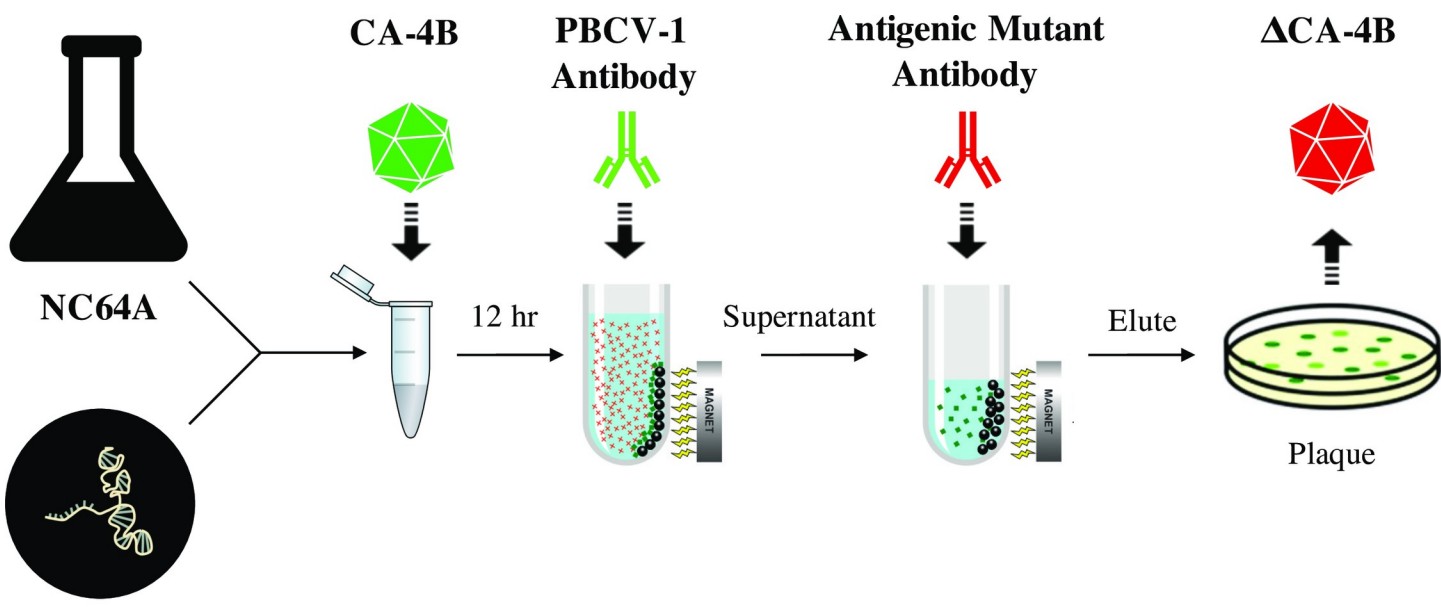

**Fig 5. Workflow of antibody selection for mutant chloroviruses.** Following incubation of macerozyme-treated NC64A cells with Cas9/sgRNA RNPs targeting the *034r* gene, cells were infected with CA-4B (MOI 5). Rabbit polyclonal antiserum (PBCV-1 antibody, green) was added to the viral-induced cell lysate to interact with wild-type viruses. Goat anti-Rabbit IgG magnetic beads were incubated with the rabbit antibody solution and then separated using magnets. The unbound viruses were collected and subsequently incubated with antibody derived from serologically distinct PBCV-1 mutants (antigenic mutant antibody, red) that have shortened surface glycans. Magnetic beads were incubated again with the antibody solution and then separated using magnets. Bound mutant viruses were eluted from the beads and plaque assayed.

subsequent nucleotide insertion at the same site) in DNA from the other–Fig 6]. This result is fully consistent with that expected from site-directed mutagenesis targeted in the PBCV-1 *a064r* homologous gene, CA-4B *034r*, by our specifically designed Cas9/sgRNA RNP complex (Figs 4 and 5). Ten additional attempts to obtain examples of gene editing with this particular Cas9/sgRNA RNP and experimental protocol failed. In addition, in all of the sequencing performed as part of the present studies which involved DNA sequencing of the 835 bp region of interest in over 50 independent viral plaques (involving over ~30,000 total nucleotides), no supposedly "spontaneous" mutations were discovered at or near the target site for Cas9/sgRNA cleavage located 3 nucleotides upstream of the PAM site (see Fig 4).

### No evidence for HDR events using ssODNs

All attempts at producing nucleotide sequence replacement by homologous DNA recombination (HDR) using ssODNs to simultaneously create a premature stop codon and remove the

WT CCTTGTATTACGATTTTATCTGG
ΔCA-4B(1) CCTTGTATTACGATTT–ATCTGG
ΔCA-4B(2) CCTTGTATTACGATTTTTATCTGG

**Fig 6. Recovery of CA-4B site-directed mutants by RNP-targeting.** Two indels (red) in *034r* from two CA-4B variants (1 and 2) were detected and sequence verified to support the possibility of site-directed mutagenesis by preassembled Cas9 protein-sgRNA RNPs in the PBCV-1 *a064r* homologous gene. PAM site, gray. Green triangle, Cas9/sgRNA cleavage site. WT, wildtype sequence.

native MscI restriction site in the *034r* gene were unsuccessful. Amplicons of *034r* DNA fragments from recovered CA-4B viral plaques after antibody selection were incubated with MscI. Virus DNA was digested into two DNA fragments that were consistent in size with a MscI-digested amplicon thus suggesting a MscI site is intact in the 38 samples tested. The conservation of the restriction site provides no evidence for ssODN-mediated HDR using the present methods.

## Discussion

Reverse genetic manipulation of NC64A genomes is currently not possible and as a result directed modification of chloroviruses has yet to be achieved. We have attempted to produce recombinant chloroviruses for many years by placing plasmids containing a virus gene (including the glycosyltransferase described in this manuscript) into host alga cells and then infecting with a chlorovirus and screening for recombinant nascent viruses as is done for making recombinant vaccinia viruses [42]. Genetic recombination is known to occur in the chloroviruses [43, 44]. Although many vectors were constructed and various ways to introduce the plasmids into the host alga were tried, recombinant viruses were never obtained. The lack of success in some of the earliest experiments was initially thought to be due to the fact that the chloroviruses produce and package virus-encoded DNA restriction endonucleases in their virions. These restriction enzymes are involved in digesting the host DNA shortly after virus infection [45]. However, despite the fact that many of the more recent attempts were conducted with virus CA-4B, which does not appear to code for a restriction endonuclease, success was not obtained.

Our results from *Agrobacterium*-mediated transformation of NC64A demonstrated its potential for being a reasonably promising transformational system to further pursue and optimize. Here, seven PCR-positive transformants were obtained, and the presence of the viral gene fragment suggests that the T-DNA was integrated into the NC64A genome; however, more extensive studies are needed to elucidate possible factors and mechanisms contributing to the loss of the *gfp-gusA* expression and hygromycin resistance. Approaches to prevent or minimize gene silencing might prove useful in maintaining the expression of newly introduced genes. Indeed, the occurance of gene silencing in NC64A is not surprising given that it contains at least 375 homologs to genes involved in gene silencing [46]. Considering the apparent problems with gene silencing following *Agrobacterium*-mediated cell transformation, we did not consider it further for delivery of genes encoding Cas9 and sgRNA. Rather, we chose to employ *in vitro* assembled Cas9/sgRNA RNP complexes as described in the text. Other vector systems and/or promoters might also need to be tested with NC64A to develop a better transformation system. Nonetheless, the finding opens the possibility of further genetic manipulation of this commercially important microalga with other genes of interest.

Consequently, with the advent of the CRISPR/Cas9 system there was hope that such problems might be overcome. There have been many advances with CRISPR/Cas9 system-based gene editing in virology including its applications in antiviral vaccine development, especially in large DNA viruses of animals. The CRISPR/Cas9 system has been successfully applied to the gene editing of many different virus species [47], mainly focusing on antiviral therapy, functional study of viral virulence factors, and the reconstitution of commonly used viral vectors for genetically engineered vaccine development. In the research reported here, we attempted to develop a transformation system to generate stable site-directed chlorovirus mutants by targeting a chlorovirus glycosyltransferase-gene that conveniently provided a distinguishable and selectable glycan phenotype. Despite our efforts to improve methods to penetrate host NC64A cells and modify CA-4B viral DNA, we were unable to produce a reliable transformation system that supported the genetic modification of chloroviruses.

The most promising results in the present study was the recovery of two CA-4B mutants that harbored hallmark, sequence-verified, Cas9/sgRNA RNP-directed indels at the site of Cas9/sgRNA cleavage. These observations suggest successful delivery into macerozyme-treated NC64A cells of preassembled Cas9/sgRNA RNPs that resulted in a frameshift mutation in the CA-4B-encoded gene *034r* of one viral progeny and a phenylalanine to leucine codon change in a separate viral progeny. Whether the two mutations observed occurred in a single NC64A cell following the rare uptake of Cas9 RNPs or in two independent cells is not possible to determine. The odds of finding in a single experiment two distinctly different indels as "spontaneous" mutations precisely located at the Cas9/sgRNA cut site in two independent viral plaques is extremely unlikely.

However, despite many trails and alternative enzyme combinations, we were unable to duplicate these results. It is unlikely that detection of the mutant virus DNA was due to erroneous sequencing given that all samples were sequenced in triplicate with consistent readings. One possible explanation for obtaining a potential RNP-mediated NHEJ event in CA-4B viral DNA is that macerozyme on a rare occasion successfully eroded the alga cell wall, promoting cell permeability, allowing the entrance of the RNP-targeting cargo. Despite targeting cell-wall polymers, macerozyme did not compromise all the host receptors sufficiently to prevent virus attachment as evidenced by routinely productive virus infections in macerozyme-treated cells. The host receptor, which is likely a carbohydrate, is uniformly present over the entire surface of the alga [48]. Once inside the infected cell, deposited viral DNA could be recognized by the gene-targeting RNPs. The lack of success of the SiC and electroporation procedures in producing mutant viruses is somewhat surprising given evidence of RNP delivery inside NC64A was greater using these delivery techniques compared to macerozyme-treated cells (Fig 3). Perhaps the fluorescently labeled cells, although in greater quantity, were either nonviable or compromised in health that prevented the desired viral DNA editing.

In one study [49], 14 different strains of *Chlorella* were challenged with cell wall-degrading enzymes and no two strains had the same pattern of inhibition. This large range of sensitivity to various enzymatic activities illustrates the wide diversity of cell wall composition amongst the *Chlorella* alga species. Our results show that in methodologies or processes using intact algal cells or residual algal biomass, enzymatic treatment can have large impacts on the permeability of the algal cell walls and may be useful in optimization of various processes.

FACS analysis of SiC whiskers-treated and electroporated NC64A cells showed ~12% and ~20% cell fluorescence, respectively, suggesting positive delivery of the RNPs inside the alga cells. We speculate that cell wall perforations either induced by mechanical force or voltage allowed Cas9/sgRNA RNP complexes brief access to the inside of the cell. In theory, once viral DNA is deposited inside the alga cell, the naked nucleic acid is available for Cas9/sgRNA binding and subsequent *034r* gene cleavage. The edited viral DNA would be subsequently replicated and packaged. Following antibody selection, we expected to recover an NHEJ indel event in CA-4B at the RNP-designated target site, however, with just two exceptions, we were unsuccessful. Although putatively mutant virus plaques were recovered by our screening techniques, sequencing results confirmed that in the vast majority of recovered viruses the wildtype domain 1 of gene *034r* was intact. Given these sequencing results, the plaques produced were likely caused by escape viruses that evaded wildtype antibody binding.

We also briefly examined the effect of parameters involving cell-penetrating peptides, such as the peptide/RNP ratio and serum addition, on peptide-mediated transfection. Properties of peptides (DNA binding stability and condensation capacity) and of peptide/RNP complexes (size and surface charge) were investigated because these are known to vary as a function of the peptide/RNP ratio. Following antibody selection, we expected to recover an NHEJ indel event in CA-4B at the Cas9/sgRNA RNP-designated target site. However, we were

unsuccessful. Although virus plaques were recovered, sequencing results confirmed the wild-type domain 1 of gene *034r* was intact. Given these sequencing results, these plaques, again, were likely caused by escape viruses that evaded wildtype antibody binding.

Broadly speaking, genetic modification of chloroviruses would provide functional insight into the unusual chlorovirus encoded proteins mentioned previously such as hyaluronan synthase, potassium ion channel protein, five polyamine biosynthetic enzymes, and as addressed here, glycosyltransferases. Furthermore, adoption of a reverse genetics system would also allow the exploration of formerly characterized proteins having potential scientific and economic benefit. For example, chlorovirus genes encode commercially important enzymes such as DNA restriction endonucleases, a DNA ligase, and they contain elements for genetically engineering other organisms. Examples include viral promoter elements that function well in both monocot and dicot higher plants, as well as bacteria [50, 51], and a translational enhancer element that functions in *Arabidopsis* [52]. Chloroviruses have some of the smallest, most primitive forms of highly complex proteins that exist in higher organisms serving as a simplified fossil template to study biochemical models for mechanistic and structural studies [53]. Therefore, the development of a successful and reproducible procedure for achieving genetic modification of chloroviruses would accelerate the exploration of microalgae and their viruses for a broader range of scientific investigations and biotechnological applications. Such an achievement would be a major step forward.

## Supporting information

**S1 Fig. *In vitro* cleavage assay.** The target *034r* locus from NC64A virus CA-4B was PCR-amplified and incubated with preassembled Cas9 and sgRNA RNP complexes *in vitro*. The complete *in vitro* cleavage of the target locus confirmed active RNP formation. PCR product (835 nucleotides) was amplified using primers upstream and downstream of domain 1. Arrowheads indicate cleaved products. L, molecular size ladder.
(TIFF)

**S1 Raw images.**
(PDF)

**S1 Dataset.**
(PDF)

## Acknowledgments

The authors want to acknowledge the contributions of their many colleagues who have been involved in many previous (unsuccessful) attempts to develop methods to produce recombinant chloroviruses. We thank Dr. Keiji Numata (RIKEN) for providing the cell penetrating peptides used in our studies.

## Author Contributions

**Conceptualization:** Donald P. Weeks, James L. Van Etten.

**Data curation:** Eric A. Noel.

**Formal analysis:** Eric A. Noel, Donald P. Weeks, James L. Van Etten.

**Funding acquisition:** Eric A. Noel, James L. Van Etten.

**Investigation:** Eric A. Noel.

**Methodology:** Eric A. Noel, Donald P. Weeks, James L. Van Etten.

**Project administration:** Donald P. Weeks, James L. Van Etten.

**Resources:** Donald P. Weeks, James L. Van Etten.

**Supervision:** Donald P. Weeks, James L. Van Etten.

**Visualization:** Donald P. Weeks, James L. Van Etten.

**Writing – original draft:** Eric A. Noel.

**Writing – review & editing:** Eric A. Noel, Donald P. Weeks, James L. Van Etten.

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
