## [Decision Letter · Decision Letter 0]

22 Jun 2021

PONE-D-21-16428

Pursuit of chlorovirus genetic transformation and CRISPR/Cas9-mediated gene editing

PLOS ONE

Dear Dr. Noel,

Thank you for submitting your manuscript to PLOS ONE. This manuscript has been reviewed by two expert reviewers in the gene editing field. Based on their evaluation and my own assessment, my decision is "revision" as it has merit but does not fully meet PLOS ONE’s publication criteria as it currently stands. Therefore, I invite you to submit a revised version of the manuscript that addresses the points raised during the review process.

Specifically, please address the two major concerns raised by the reviewers:

1. Frequency of obtained gene edited events: The reviewer 1 is convinced that the obtained mutations are indeed achieved by Cas9-mediated gene editing but more information such as efficiency and target sequences should be provided. The reviewer 2 raises concerns whether the gene edited events could be obtained by natural mutations. Thus, please provide information to exclude the occurrence of natural mutation or explain why this is indeed caused by gene editing.

2. Writing style and format of the article: Although the results of this manuscript did not achieve the original goal by establishing the reproducible gene editing method, there are still many positive aspects reported in this manuscript. I agree with the reviewer 1 that the abstract should describe more results obtained during the method development than focusing on the negative results. As this is a manuscript for a method development, some parts of methods can be described as part of results. 

We look forward to receiving your revised manuscript.

Kind regards,

Erh-Min Lai, Ph.D.

Academic Editor

PLOS ONE

Journal Requirements:

4. Please amend your manuscript to include your abstract after the title page.

Reviewers' comments:

Reviewer's Responses to Questions

**Comments to the Author**

1. Is the manuscript technically sound, and do the data support the conclusions?

Reviewer #1: Partly

Reviewer #2: No

2. Has the statistical analysis been performed appropriately and rigorously? 

Reviewer #1: N/A

Reviewer #2: No

3. Have the authors made all data underlying the findings in their manuscript fully available?

Reviewer #1: Yes

Reviewer #2: No

4. Is the manuscript presented in an intelligible fashion and written in standard English?

Reviewer #1: Yes

Reviewer #2: No

5. Review Comments to the Author

Reviewer #1: Engineering DNA viruses of green algae could be important biotechnology to solve the environmental problem such as algae bloom. Here ,the attempt is to transform CRISPR/Cas9 into the algal host Chlorella variabilis NC64A and infected with Chloroviruses. Theoretically, genome edited Chlorovirus can be obtained. The pre-requested technique is the ability to transform Chlorella variabilis NC64A. The authors tried several transformation methods including electroporation of RNP, transformation after cell-wall digestion, silicon carbide whiskers, cell-penetrating peptides transformation of RNP and Agrobacterium transformation of DNA. Most of these transformation methods were unsuccessful to obtain mutated virus, however, two mutated virus were identified from macerozyme-digested algae. This result demonstrated CRISPR/Cas9-mediated gene editing of chlororivrus genome is possible.

This article has clear introduction of background knowledge. The authors also properly discussed the results they obtained. Overall, the experimental design is logical to achieve their goal. Most importantly, they obtained two mutated viruses with different indels, indicating this experiment is success in certain level.

Notably, I think the authors did not fully aware of the differences between a research article and an article of methodology. Two issues are likely related to this problem: 1-“Methods” part is a mixture with “results and methods”; 2-unable to duplicate these results of edited chloroviruses.

Current “method” part is very difficult to read, I am sorry to say. For a research article published on PLOS ONE, the procedure of experimental designs and first figure are put at “Method” part before “Results” section could be common. But for a methodology research, development is part of results. Please consider to put only “methods” that mostly followed previous references with minor modifications in the “methods”.

Second, the authors used a negative way to describe their result: “they were unable to duplicate these results and therefore unable to achieve a reliable system to genetically edit chloroviruses” in the abstract and most of times, besides in the discussion. However, performing gene editing in non-model organisms is not an easy task, it is even more challenge to edit viral genome of this non-model organism. I still think this research has its breakthrough. For this, I would suggest the authors try to describe this result from a positive aspect. Also, the results from these two mutated viruses could be more solid. Here are some suggestions:

- These mutants may come from rare events, but how rare it is? How many plagues were sequencing eventually? How many were from macerozyme digestion? How many were from rest of other enzymes?

- Fig 3C, please also show chromatograph picture of Sanger sequencing and mark these indel and PAM information.

- Besides therapeutic purpose such as preventing virus infection, how many success cases of manipulating viral genome by CRISPR?

Other suggestions:

Fig 1 is experimental design should not only appear only at introduction.

Table 1. Would be better if you can show the quantification results of cell growth inhibition by enzyme.

Line 113

Fig 1. The cleavage site (green triangle) should be between TTTT and ATCTGG, three nucleotides away from PAM, also Fig 2C. This actually increases the reliability of the result that mutations were indeed came from Cas9 cleavage, the 1-nt deletion and 1-nt insertion are all adjacent to the cleavage site, two very common insertion and deletion.

Line 132; 217

“sgRNA” to “sgRNA or crRNA”. Cas9 uses an artificial sgRNA, Cpf1 uses its native crRNA. They cannot use the same sgRNA. The target site of Cpf1 is TTTV, followed by protospacer target sites at downstream of TTTV PAM. What is the target sequence of Cpf1?

Line 490 “we were unable to produce a reliable transformation system”

The description here is not complete wrong but can be more positive. I tried to image the difficulty of obtaining the mutated virus. Is that correct that mutated viral genome may not be enveloped into mutated envelop? CRISPR/Cas9 is not an efficient nuclease at short-time window in many organisms. If only a small proportion of viral genome is mutated during their duplication in host cell, only an even smaller proportion of mutated viral genome is enveloped into mutated viral capsid, is that possible?

Line 383 “Two independent plaques”

Is that possible these two virus mutants came from the same host cell?

Line 391 “Many attempts to obtain additional examples of gene editing with this particular Cas9/sgRNA RNP failed.”

Please specify. How many batches of experiment were performed? How many plagues were analyzed? How many plagues are wildtype 034r gene?

In Fig 5C, resolution of plate picture is too low to show colonies.

Fig 6.” imaged through green, red, and GFP channels (from left to right).”

The description of respective images is not clear. Frankly speaking, I am highly suspicious to the fluorescent signals of GFP. These signals overlapped nearly completely with Red channel (is this autofluorescence signal, from chloroplast?). Unless the GFP-GUS proteins have chloroplast translocation signal peptide, otherwise it is unlikely overlapped with chloroplast. In plant cells, GFP can be observed in cytosol and nucleus, but not in chloroplast and vacuole. Also, GUS usually is a much better reporter than GFP. Besides, please confirm white light image. If all cells here are all with GFP signals, the transformation rate is unreasonably high. The presence of the gfp-gusA gene PCR could come from agrobacterium.

Reviewer #2: The authors tried to establish reverse-genetics system in chlorovirus and obtained mutants in which site-directed mutation was likely to be introduced. Also, they used Agrobacterium-mediated transformation system and succussed to deliver a vector into host cell of chlorovirus. Their work is important, and some promising results were obtained. However, the article has some fatal problems. The data provided in the article is insufficient to support the conclusion. Some data is not closely related to the main topic. The article structure is not well organized. Overall, additional information and large reconstruction are required for publication. Major concerns are below.

1. The two mutants obtained was likely to result from Cas9 cleavage and following NHEJ. However, the possibility of natural mutation cannot be excluded based on the provided data. The authors used two types of antibodies for wildtype and mutant surface structure, which may select the viruses with natural mutation in the glycosyltransferase. To exclude the possibility, the authors should demonstrate the mutation was introduced only in the targeted site at significantly higher ratio than natural mutation. Full length of 034r gene should be sequenced from all antibody-selected viruses and mutation rate in the gene should be calculated.

2. Agrobacterium-mediated transformation system is promising, but not closely related to the reverse-genetics system in chlorovirus. If the authors had used the system for gene-editing, it could be included. Otherwise, this part should be removed from this article.

Below are minor comments.

1. In method section, order of information should be consistent with the order of presented results. Used chemicals should be summarized in early part of the section.

2. As the trial of homologous recombination failed, this part can be reduced. One sentence with “data not shown” is enough.

3. Table 1: Growth inhibition is ambiguous and subjective. The data should be presented in more quantitative and objective way.

4. Figure 2: This image includes experimental scheme and results. They should be separated.

5. Line 472: This paragraph is a kind of history. It is interesting but should be more organized for scientific paper.

6. Experimental procedure is fairly overlapped in all sections. Small overlap should be allowed, as the article mainly described methods, but it is too much.

6. PLOS authors have the option to publish the peer review history of their article (what does this mean?). If published, this will include your full peer review and any attached files.

Reviewer #1: No

Reviewer #2: No

---

## [Author Response · Author response to Decision Letter 0]

15 Jul 2021

These items are addressed in our Responses to Reviewers document.

---

## [Decision Letter · Decision Letter 1]

16 Aug 2021

PONE-D-21-16428R1

Pursuit of chlorovirus genetic transformation and CRISPR/Cas9-mediated gene editing

PLOS ONE

Dear Dr. Noel,

Thank you for submitting your manuscript to PLOS ONE. The revised manuscript has been re-evaluated by two reviewers who reviewed previous version. The reviewer 1 is mostly satisfied with the revision but also provides some comments for further improvement of the manuscript. Reviewer 2 remains several concerns that have not been fully addressed. After careful consideration, we feel that it has merit but some minor revisions are required prior to publication. Therefore, we invite you to submit a revised version of the manuscript that addresses the specific points indicated below and other comments if applicable.

1. As indicated by reviewer 1, please confirm the identity of gfp-gusA reporter vector and name consistency and accuracy (e.g. gfp-gusA or GFP-GUS reporter and using italics for Agrobacterium tumefaciens throughout the paper)

2. Please provide the evidence that GFP signal is derived from transformation event (Figure 3). Both reviewers concern the GFP signal could be autofluorescence considering that all cells shown have signals while no GUS activity could be detected. Please provide the image of negative controls and discuss the reason of no GUS signals. If not, please remove this figure.

We look forward to receiving your revised manuscript.

Kind regards,

Erh-Min Lai, Ph.D.

Academic Editor

PLOS ONE

Journal Requirements:

Reviewers' comments:

Reviewer's Responses to Questions

**Comments to the Author**

1. If the authors have adequately addressed your comments raised in a previous round of review and you feel that this manuscript is now acceptable for publication, you may indicate that here to bypass the “Comments to the Author” section, enter your conflict of interest statement in the “Confidential to Editor” section, and submit your "Accept" recommendation.

Reviewer #1: All comments have been addressed

Reviewer #2: (No Response)

2. Is the manuscript technically sound, and do the data support the conclusions?

Reviewer #1: Yes

Reviewer #2: Partly

3. Has the statistical analysis been performed appropriately and rigorously? 

Reviewer #1: N/A

Reviewer #2: N/A

4. Have the authors made all data underlying the findings in their manuscript fully available?

Reviewer #1: Yes

Reviewer #2: No

5. Is the manuscript presented in an intelligible fashion and written in standard English?

Reviewer #1: Yes

Reviewer #2: No

6. Review Comments to the Author

Reviewer #1: The authors developed two transformation system: agrobacterium-mediated transformation and electroporation to transform algae Chlorella variabilis NC64A. The electroporation of in vitro synthesized CRISPR/Cas9 RNP complex together with Chloroviruses infection successfully generate two mutated Chloroviruses adjacent to the cleavage site of viral genome.

I think the current version description is reader friendly. The data support their main conclusions.

Line390. I think it would be better if the authors can make a description why agrobacterium-transformation was not used for gene editing in this study due to the concern of gene silencing.

Line36-37 In abstract “However, we were unable to duplicate these results and therefore unable to achieve a reliable system to genetically edit chloroviruses. Nonetheless…”. I guess the authors want to deliver a message that current method required further improvement. However, I think this point of view is fully discussed in the text already. “2 mutated viruses were obtained in total 56 plague” or “2 mutated viruses were obtained in only one of five replicates” are both correct description but give opposite impression to the readers

Some paragraphs are not changed to the new orders accordingly.

Paragraph Line 156-161: “Cell wall-degrading enzymes” could be moved after Line 204.

Paragraph Line 582-591 “Finally, our results from Agrobacterium-mediated…” could be moved after Line 525.

Result HDR analysis, paragraph line392-399

I think that although it is not necessary to show MscI digestion result in gel, but to evaluate a transformation or gene editing method, the analyzed sample numbers are critical. The MscI digestion to confirm HDR events might be fail to proof any success event, I think if the authors can show how many samples were confirmed in this part, we could have a better idea how far it is to reach successful HDR. Beside MscI digestion, the samples were sequenced should be included as well. If all MscI analyzed samples were sequenced, it is also fine.

Minor point:

Line370-371 gfp-gusA genes in pCAMBIA1303

As far as I checked, pCAMBIA1303 contain gus-gfp. Please confirm that pCAMBIA1303 and pCAMBIA1304 mentioned in this article are all correct.

Line 400-412 Fig 4 in this paragraph is Fig 1.

Line 440-441 the expected 600 bp and 235 bp

After the correction of cleavage site, why the size of these two fragments are unchanged?

Line 468-469 “a nucleotide substitution…a nucleotide substitution…”

To be more precise, I would suggest “one nucleotide deletion…one nucleotide insertion…”.

Line539-540 Cas9/sgRNA RNP-directed indels precisely at the site of Cas9/sgRNA cleavage.

The description is very likely to be true but a bit too strong to use “precisely”. Unfortunately, because the insertion or deletion are located at the original TTTT sequence, it is indistinguishable the insertion/deletion are occur at the last T next to cleavage site or are occur at other positions.

Please use consistent name and style of gfp-gusA for gene or GFP-GUS reporter for protein.

e.g. Line162: gfp:gusA fusion reporter (:?); Line367: GFP-GUS reporter gene (gene); Line393: gfp-gusA reporter(no italic).

Cas9/gRNA or Cas9/sgRNA are both fine. But I would suggest to use consistent name in an article.

Similar issue, CRISPR/Cas9 in reference list are all showing as Crispr/Cas9? I checked some PLOS ONE articles and their reference list still used CRISPR/Cas9 or CRISPR/CAS9. I don’t think using CRISPR violate the style of PLOS ONE reference format

Fig 5. The sequences of virus and ODN could be aligned better.

Fig 5. The marker for sgRNA, gray color is correct but line art is one-nucleotide longer.

Reviewer #2: The manuscript was modified with some improvements. However, there are still a lot of major problems, including those I missed in the last time. The manuscripts must be largely improved to fulfill the minimum requirements for scientific publication.

Major points:

Line 381–383: GFP expression is not fully supported by the provided data, considering the following statement. It is strange only GFP was expressed at protein level and GUS was not. The result implies that GFP observed in Figure 3 is autofluorescence or artificial. To validate GFP expression, authors must provide the image of negative control.

Line 410: In the manuscripts, inhibition effect is described with two levels, “the greatest negative effect” or “minor growth inhibition”. This description is not consistent with three-level description in the table legend, “complete”, “moderate”, and “slight”. They must be consistent. Also, if the effect is judging from the size of spots, actual size must be provided as numerical number. At least, photo image must be provided as a supplemental figure.

Line 476–482: Total number of plaques sequenced in each attempt must be precisely presented as Table. Also, the number of mutations found in those sequences must be provided.

Line 478–482: Basically, it is impossible to discriminate NHEJ-induced mutations from natural mutation. Therefore, it is important to show the mutation rate in Cas9-cleavage site is higher than other sites, and authors’ description, ‘no supposedly “spontaneous” mutations were discovered at or near the target site for Cas9/sgRNA cleavage located 3 nucleotides upstream of the PAM site’ is not response to the suspicion. The authors must provide the number of mutations found in target and non-target sites in all the attempts. Also, all the sequence must be provided at least for the first experiment as a supplemental figure.

Line 579–580: Judging from the manuscripts, only a part of 034r gene was sequenced. How do they conclude that the wildtype 034r gene was intact?

Line 590–591: If the authors think the Agrobacterium-mediated method is promising, why did they give up the method and move to enzymatic method?

Minor points:

Overall, they should use more scientific terms and phrases, and provide “necessary and sufficient” information with clear separation into Introduction, Materials and Method, Results, and Discussion.

Line 17: The sentence should start from “Genetic and molecular modifications…”

Line 26: Does T-DNA mean the T-DNA region in Ti plasmid?

Line 27: “and gene silencing…of this system” should be removed.

Line 27: The sentence should start from “To develop…”

Line 68–99: These three paragraphs have redundant and irrelevant information, so they should be reorganized.

Line 94–99: This information should be moved to results or methods.

Line 123: “(caused by Cas9/gRNA-directed gene editing)” should be removed, because this information is misleading. The authors’ method cannot discriminate Cas9-induced mutation from natural mutation.

Line 108–124: This information should be moved to results or methods.

Line 125–146: Information in this paragraph should be more generalized to explain the rational of CRIPSR-Cas9 system.

Line 149–152: In description of chemical concentration, hydrated water should be removed.

Line 164: What is CaMV?

Line 181: “accessed” should be changed to “counted”.

Line 199–200: Information on the place of microscope should be separated from that on manufactures.

Line 244: Show exact sequence of the primers.

Line 240: What is GENEWIZ? Is it company or machine for sanger sequence?

Line 249: Agar concentration should be shown as percentage.

Line 257: What is the purpose of a solution?

Line 259: A word, “previously” should be removed.

Line 269: A phrase “by NHEJ” should be removed. PCR and sequence cannot be discriminate mutations by NHEJ from those by other events.

Line 270: A word “first” should be removed.

Line 291: A phrase “by NHEJ” should be removed. PCR and sequence cannot be discriminate mutations by NHEJ from those by other events.

Line 301: A phrase “by NHEJ” should be removed. PCR and sequence cannot be discriminate mutations by NHEJ from those by other events.

Line 314–316: This is a result.

Line 323: A phrase “by NHEJ” should be removed. PCR and sequence cannot be discriminate mutations by NHEJ from those by other events.

Line 327: “tumefaciens” should be italic.

Line 332–333: The growth seemed to be inhibited at 100 mg/L in Figure 1A.

Line 343–344: The Y-axis label of B and C should be the same.

Line 347–351: Schematic image of the plasmid construction should be provided.

Line 370: Why are the sequences of hpt and gfp-gusA different from original sequence? Sequence of the original plasmid constructed in this study must be checked.

Line 378–379: Figure 2C is not required.

Line 382: Fluorescent microscopy confirms expression of GFP protein.

Line 398: The bar should be precisely 5 µm.

Line 405: A word “greatest” should be modified.

Line 420–423: These percentage should be presented in Figure 4.

Line 439–441: The intensity of cleaved band is too faint. Authors must present clearer image for this experiment.

Line 467: Show exact sequence of the primers.

Line 472–476: This part should be moved to discussion.

Line 482–484: This part should be moved to discussion.

Line 498–499: “site-directed mutagenesis by preassembled Cas9 protein-sgRNA RNPs” cannot be verified only by the sanger sequence results.

Line 505: A word “also” should be removed.

Line 508: A phrase should be “MscI site is intact”

7. PLOS authors have the option to publish the peer review history of their article (what does this mean?). If published, this will include your full peer review and any attached files.

Reviewer #1: No

Reviewer #2: No

---

## [Author Response · Author response to Decision Letter 1]

15 Sep 2021

These items are addressed in our Responses to Reviewers document.

---

## [Editor Report · Decision Letter 2]

29 Sep 2021

Pursuit of chlorovirus genetic transformation and CRISPR/Cas9-mediated gene editing

PONE-D-21-16428R2

Dear Dr. Noel,

We’re pleased to inform you that your manuscript has been judged scientifically suitable for publication and will be formally accepted for publication once it meets all outstanding technical requirements.

Kind regards,

Erh-Min Lai, Ph.D.

Academic Editor

PLOS ONE
---

## [Editor Report · Acceptance letter]

12 Oct 2021

PONE-D-21-16428R2 

Pursuit of chlorovirus genetic transformation and CRISPR/Cas9-mediated gene editing 

Dear Dr. Noel:

I'm pleased to inform you that your manuscript has been deemed suitable for publication in PLOS ONE. Congratulations! Your manuscript is now with our production department. 

Kind regards, 

on behalf of

Dr. Erh-Min Lai 

Academic Editor

PLOS ONE